# Evaluating patient factors, operative management and postoperative outcomes in trauma laparotomy patients worldwide: a protocol for a global observational multicentre trauma study

Michael F Bath [1], Katharina Kohler [1,2] Laura Hobbs,[1,3]
Brandon George Smith [1], David J Clark [4], Arthur Kwizera [5],
Zane Perkins,[6,7] Max Marsden,[6,8] Ross Davenport,[6,7] Justin Davies,[9,10]
Joachim Amoako,[11,12] Ramani Moonesinghe [13] Thomas Weiser [14],
Andy J M Leather,[15] Timothy Hardcastle,[16,17] Ravi Naidoo,[18] Yannick Nördin,[19]
Andrew Conway Morris [20], Kokila Lakhoo,[21] Peter John Hutchinson [4,22]
Tom Bashford [1,2]

For numbered affiliations see end of article.

**Correspondence to**
Dr Michael F Bath;
mb2583@cam.ac.uk

## ABSTRACT

**Introduction** Trauma contributes to the greatest loss of disability-adjusted life-years for adolescents and young adults worldwide. In the context of global abdominal trauma, the trauma laparotomy is the most commonly performed operation. Variation likely exists in how these patients are managed and their subsequent outcomes, yet very little global data on the topic currently exists. The objective of the GOAL-Trauma study is to evaluate both patient and injury factors for those undergoing trauma laparotomy, their clinical management and postoperative outcomes.

**Methods** We describe a planned prospective multicentre observational cohort study of patients undergoing trauma laparotomy. We will include patients of all ages who present to hospital with a blunt or penetrating injury and undergo a trauma laparotomy within 5 days of presentation to the treating centre. The study will collect system, patient, process and outcome data, following patients up until 30 days postoperatively (or until discharge or death, whichever is first). Our sample size calculation suggests we will need to recruit 552 patients from approximately 150 recruiting centres.

**Discussion** The GOAL-Trauma study will provide a global snapshot of the current management and outcomes for patients undergoing a trauma laparotomy. It will also provide insight into the variation seen in the time delays for receiving care, the disease and patient factors present, and patient outcomes. For current standards of trauma care to be improved worldwide, a greater understanding of the current state of trauma laparotomy care is paramount if appropriate interventions and targets are to be identified and implemented.

## STRENGTHS AND LIMITATIONS OF THIS STUDY

⇒ The GOAL-Trauma Study will provide a global snapshot of current standards in care for patients undergoing trauma laparotomy, from time of injury to time of discharge.

⇒ The data set includes aspects of patient and injury factors that will provide improved understanding in the barriers and target areas to improve overall care and outcomes for trauma patients.

⇒ A global panel of trauma experts has developed with protocol, with significant involvement from patient and public involvement groups.

⇒ This study is purely observational, therefore, no new treatment or management options are being analysed.

## INTRODUCTION

Trauma accounts for approximately 9% of all global deaths[1] and contributes the greatest loss of disability-adjusted life-years for adolescents and young adults worldwide.[2] In the context of global abdominal trauma, both blunt and penetrating, the trauma laparotomy is the most commonly performed operation; it can be used both as a means to access identified injuries to organs within the abdomen and pelvis, such as life-threatening bleeding from vessels or solid organs, or contamination from hollow viscus injuries, or for exploratory diagnostic reasons, where cross-sectional imaging may not be easily feasible or available. While mortality rates following a trauma laparotomy are substantial, ranging between 8% and 40%,[3–8] in recent years the role of damage control resuscitation has transformed trauma care, with significant reduction in mortality and morbidity to trauma patients being

observed.[9–11] However, the standard of care and variation in practice occurring globally for patients undergoing a trauma laparotomy remain relatively unknown.

Comprehensive trauma care is a complex system that includes multiple components, from the first response and prehospital care elements, to definitive facility-based care, and then final rehabilitation and recovery, all while interacting with multiple healthcare services and regional infrastructures.[12] As such, ensuring optimal trauma care provision can prove difficult, especially where health systems are fragile or under-resourced. A trauma laparotomy is a key procedure within emergency surgical care,[13] yet if global trauma care is to be improved and optimised, high-quality and nuanced data of patients accessing the trauma system is needed, from across the whole trauma care pathway.

Variation likely exists worldwide in the management and outcomes of patients following trauma laparotomy,[14] however at present limited data exists. While similar post-operative outcomes have been observed in major trauma centres in a high-income setting,[3] this is likely not representation on the global scale, alongside varying injury patterns.[15] Indeed, variations in the epidemiology and management of traumatic brain injury have been noted on the international level.[16] Ensuring adequate trauma laparotomy care globally is essential if trauma outcomes are to be improved, however a greater understanding of the current state of trauma laparotomy care is paramount if appropriate interventions and targets are to be identified.

The objective of this study is to evaluate both patient and injury factors for those undergoing trauma laparotomy, their clinical management and postoperative outcomes.

## METHODS

### Aims

The primary aim of this study is to describe global 30-day postoperative mortality rates for patients undergoing a trauma laparotomy.

The secondary aims of the study are as follows:
1. To describe the epidemiological characteristics (demographics, injury characteristics, baseline clinical characteristics and surgical casemix) for patients undergoing a trauma laparotomy.
2. To describe the preoperative, perioperative and postoperative processes of care for patients undergoing a trauma laparotomy.

### Study design

We will conduct a prospective multicentre observational cohort study of patients undergoing a trauma laparotomy. We will recruit centres through pre-existing research networks, using a snowballing technique to expand registration of centres. We will enrol eligible patients undergoing emergency trauma laparotomy over a consecutive 30-day period at individual centres during the study dates,

with patients followed until discharge, death or 30 days postoperatively, whichever comes first. All data will be submitted to a central study team by 30 days after the end of their respective data study period.

Any hospital worldwide that performs emergency trauma surgery will be eligible to participate, including both trauma centres and trauma units; a minimum of one case must be submitted by the centre during the 30-day study period to be eligible. Each centre's team will be composed of a local study lead, with maximum three members of the local study team for each data collection period. Independent data validators will also be required for select centres. The data collection periods will run between 1 April 2024 and 31 December 2024, with final 30-day follow-up for the final patients expected to be completed on 30 January 2025.

The study will be overseen and managed by the study's steering committee, with external advice provided by the study's advisory committee (see online supplemental material), all involved in the study design and protocol write-up.

### Patient criteria

We will include patients of all ages who present to hospital with a blunt or penetrating injury and undergo a trauma laparotomy within 5 days of presentation to the treating centre (day 0 being time of presentation). A patient will be included for final analysis if >70% of the data points required have been recorded.

Exclusion criteria are as follows:
- Patients undergoing a laparotomy ≥5 days (ie, 120 hours) since presentation to the treating centre.
- Any relook laparotomy, including transfers from another centre for further and/or definitive surgery.
- Any laparoscopic (including laparoscopic converted to open), robotic, or image-guided procedures.
- Patients who have been recently discharged from any hospital (including for non-trauma-related admissions) and have represented within 30 days of discharge.

### Data collection

We aim to collect system, patient, process and outcome data. A data set will be collected on all patients undergoing a trauma laparotomy within the inclusion period (see online supplemental table 1). The included data fields were based on work by similar studies and refined through iterative consultation with a global interdisciplinary consortium of clinicians involved in trauma care.

Data will be collected through access to patient records only at each centre, performed by members of the local study team. The patient will not be contacted directly in any capacity during their inpatient stay or after the study and no direct involvement in patient care will occur. Data will be collected directly onto a well-established secure web-based system, REDCap cloud (or recorded temporarily onto a hard copy data collection form and uploaded to REDCap at a later date).

 Bath MF, *et al. BMJ Open* 2024;**14**:e083135. doi:10.1136/bmjopen-2023-083135

A local data validator (independent of the local study team) will be appointed at select participating centres to assess the data accuracy and ascertainment. They will independently assess case ascertainment and collect data on two variables, namely the date and nature of the operation. Each centre's local lead will also be asked to complete a centre survey, to provide more background on their specific hospital settings, infrastructure and capacity. A pilot study was run prior to the main data collection period at multiple centres globally to ensure feasibility in the methodology across multiple settings.

The data collected will include a pragmatic set of variables that allow the proposed research to be conducted, but minimise both risk of identification and exclude extraneous possibly sensitive medical information. Local requisite ethics and approvals will be applied for and be in place before any data collection occurs. The patient consent procedure will depend on the local and national guidance for the collaborating hospitals; specific guidance will be provided to collaborators on consenting if necessary and, where required, written consent forms will be obtained for patient inclusion in the study.

Access to the main data for analysis will be limited to named researchers from the steering committee, however, collaborators at each centre will be able to access their own institutions data during the study period.

## Statistical analysis

A formal statistical analysis plan will be produced. However, in brief, participating centres will be stratified based on their country into groups based on their Human Development Index (HDI) rank, with 30-day mortality then reported for each HDI group. Further analyses will include stratification against other economic and health indices, and multivariable logistic modelling will be used to investigate primary and secondary study aims.

As there are a lack of data on trauma laparotomy globally, performing an accurate sample size calculation is not feasible. Data suggest that 30-day mortality rates following trauma laparotomy in high-income countries (HICs) are around 9%[3] and in low-income and middle-income countries (LMICs) are around 17%[17–19]; using an alpha value=0.05 and power=80%, to show a difference between HIC and LMIC settings, a total of 552 patients are required to be recruited to the study. Based on data from other observational studies,[3 7 8] we would expect over a 30-day observation period each centre to commit around 3–5 patients to the study. Therefore, with around 150 recruiting centres from our pre-existing research network, we would expect to meet the sample size required.

Missing values are to be expected in a study of this kind and will potentially affect certain centres more than others. To allow for potential imputation without introducing significant bias, we will include patients with ≥70% completeness. Multiple imputation methods will be used for the remaining missing data.

## Ethical approval

The study has received ethics approval from the Cambridge Psychology Research Ethics Committee PRE.2023119). According to the UK National Health Service Health Research Authority tool (https://www.hra-decisiontools.org.uk/research/), the study can be considered locally as a clinical evaluation study in the UK. Outside of the UK, each centre's local team will be responsible for obtaining the necessary local approvals in line with their hospital and/or national regulations. Collaborators will need to confirm local approval has been granted prior to of uploading patient data.

The study has been pre-registered to ClinicalTrials.gov (NCT06180668) and will be reported following the Strengthening the Reporting of Observational Studies in Epidemiology statement guidelines and checklists.[20] The sponsor of the study will be the University of Cambridge, with support provided from a number of collaborators. Once complete, data and conclusions from the study will be disseminated through the research network to all collaborators, with all work from the study planned to be published in open-access journals. As an observational study, no future amendments to the protocol are planned.

## Patient and public involvement

Patient and public involvement (PPI) groups were involved in the development of the study protocol and key participant literature. During the protocol development stage, a PPI discussion group was convened, involving with people who have specific lived experiences relevant to the project; feedback obtained from this session directly influenced aspects of methodology and proposed outcomes for the study. A separate PPI group further reviewed the proposed patient-focused literature associated with the study, providing comments and edits to this work that were implemented in the eventual patient information sheets produced.

## DISCUSSION

We present the protocol for a global observational multicentre study, the GOAL-Trauma study, which aims to evaluate the patient and injury factors in those undergoing a trauma laparotomy and to assess any variation observed in outcomes occurring globally. This study will provide a global snapshot of the current 30-day mortality rates following trauma laparotomy, as well as insight into any variation observed in the time delays for receiving trauma care, the disease and patient factors present and patient outcomes.

There remains a relative paucity of evidence regarding global outcomes following trauma laparotomy. Much of the current data on trauma laparotomy derives from the military setting, and the remaining civilian-level data currently published predominantly comes from high-income settings. Without high-quality data available, ensuring a greater awareness and development towards global trauma strategies is challenging. Trauma

laparotomy is a vital procedure in trauma care, therefore this study will provide key insights into trauma management.

Trauma care is complex, with heterogeneous patient groups and disease presentations, set within varying macrohealthcare and microclinical requirements.[21] This study will collect data across the patient journey, allowing key parts and limitations within this complex journey of trauma care to be identified and unpicked. However, data collection methods can often be challenging in LMIC settings, as robust numerical data may not be readily available, limiting conventional analysis techniques. We will use the Abbreviated Injury Scale (AIS) to categorise the wide spectrum and severity of traumatic injuries sustained,[22] with the individual AIS scores further used to calculate an Injury Severity Score for each patient, quantifying overall injury burden and will be adjusted for in the final data analysis.

As with any trauma study, there remains a large selection bias, as there will be many unaccounted mortalities from trauma that did not make it to hospital, did not make it into theatre, or deemed not appropriate for surgery, therefore not included in the study. We also suspect there may be a relatively high amount of missing data, however we will attempt to adjust for this with multiple imputation methods. Of further note, we are not including patients who have undergone any laparoscopic procedure (including laparoscopic converted to open), alongside any robotic procedures or interventional radiology procedures, to ensure the core data set provides the clearest picture on trauma provision and pathways globally.

The GOAL-Trauma study will provide high-quality observational multicentre data on an important topic in trauma care globally, using the trauma laparotomy as a surrogate marker. Identifying the key aspects of patient and injury factors in this patient cohort, both prehospital and in-hospital, will ensure the primary data required to understand barriers and key target areas to improve overall care and outcomes.

**Author affiliations**

[1] International Health Systems Group, Department of Engineering, University of Cambridge, Cambridge, UK
[2] Department of Anaesthesia, Cambridge University Hospitals NHS Foundation Trust, Cambridge, UK
[3] Department of Anaesthesia, East and North Hertfordshire NHS Trust, Stevenage, UK
[4] Division of Neurosurgery, Department of Clinical Neurosciences, University of Cambridge, Cambridge, UK
[5] Department of Anesthesia, Makerere University, Kampala, Uganda
[6] Centre for Trauma Sciences, Blizard Institute, Queen Mary University of London, London, UK
[7] Major Trauma Service, Royal London Hospital, Barts Health NHS Trust, London, UK
[8] Academic Department of Military Surgery and Trauma, Research and Clinical Innovation, Defence Medical Services, Birmingham, UK
[9] Cambridge Colorectal Unit, Addenbrooke's Hospital, Cambridge University Hospitals NHS Foundation Trust, Cambridge, UK
[10] Department of Surgery, University of Cambridge, Cambridge, UK
[11] Department of Surgery, Korle Bu Teaching Hospital, Accra, Ghana
[12] University of Ghana Medical School, Accra, Ghana
[13] National Clinical Director for Critical and Perioperative Care, NHS England, London, UK
[14] Department of Surgery, Stanford University, Palo Alto, California, USA
[15] School of Life Course and Population Sciences, King's College London, London, UK
[16] Department of Surgical Sciences, Mandela School of Medicine (NRMSM), University of KwaZulu-Natal, Durban, South Africa
[17] Trauma and Burns Unit, Inkosi Albert Luthuli Central Hospital, KwaZulu-Natal Department of Health, Durban, South Africa
[18] Department of Surgery, Ngwelezana Hospital, Empangeni, South Africa
[19] Emergency Medical Care System (SAMU), Jalisco State, Mexico
[20] Division of Anaesthesia, Department of Medicine, University of Cambridge, Cambridge, UK
[21] Department of Paediatric Surgery, University of Oxford, Oxford, UK
[22] NIHR Global Health Research Group on Acquired Brain and Spine Injury, Cambridge, UK

**Contributors** MFB and TB were involved in the conception of article. MFB, KK, LH, BGS, DJC, AK, ZP, MM, RD, JD, JA, SRM, TW, AJML, TH, RN, YN, ACM, KL, PJH and TB were all involved in the design and planning of the study, and in the drafting of the article. TB reviewed and finalised the article.

**Funding** University of Cambridge

**Competing interests** None declared.

**Patient and public involvement** Patients and/or the public were involved in the design, or conduct, or reporting, or dissemination plans of this research. Refer to the Methods section for further details.

**Patient consent for publication** Not applicable.

**Provenance and peer review** Not commissioned; externally peer reviewed.

**ORCID iDs**
Michael F Bath http://orcid.org/0000-0003-1879-1093
Katharina Kohler http://orcid.org/0000-0003-1919-0193
Brandon George Smith http://orcid.org/0000-0001-8471-1368
David J Clark http://orcid.org/0000-0002-4500-3892
Arthur Kwizera http://orcid.org/0000-0002-7025-0465
Ramani Moonesinghe http://orcid.org/0000-0002-6730-5824
Thomas Weiser http://orcid.org/0000-0002-3118-3888
Andrew Conway Morris http://orcid.org/0000-0002-3211-3216
Peter John Hutchinson http://orcid.org/0000-0002-2796-1835
Tom Bashford http://orcid.org/0000-0003-0228-9779

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
