## [Reviewer comments · BMJ Open]

ARTICLE DETAILS

TITLE (PROVISIONAL)	Evaluating Patient Factors, Operative Management, and Post-Operative Outcomes in Trauma Laparotomy Patients Worldwide - A Protocol for a Global Observational Multi-Centre Trauma Study
AUTHORS	Bath, Michael; Kohler, Katharina; Hobbs, Laura; Smith, Brandon; Clark, David; Kwizera, Arthur; Perkins, Zane; Marsden, Max; Davenport, Ross; Davies, Justin; Amoako, Joachim; Moonesinghe, S. Ramani; Weiser, Thomas; Leather, Andy; Hardcastle, Timothy; Naidoo, Ravi; Nördin, Yannick; Conway Morris, Andrew; Lakhoo, Kokila; Hutchinson, Peter; Bashford, Tom

VERSION 1 – REVIEW

REVIEWER	Lin, Heng-Fu Yuan Ze University, Graduate institute of medicine
REVIEW RETURNED	03-Jan-2024

GENERAL COMMENTS	Although the design of the study is great and the proposed problem about the outcomes of trauma laparotomy is important. I still have concerns about patient selection of the study. This study tries to analyze the outcomes of patients undergoing laparotomy for abdominal trauma either by penetrating or blunt mechanism, and the impact of higher incidence of associated injuries for blunt abdominal trauma might be a confounding factor. I know it's not easy to incorporate as many as 582 patients undergoing laparotomy for trauma in a 30-day period, but recruiting patients by different trauma mechanisms might be a problem for further analysis of data. Could the study designers give more explanations about the rationale about patient selection of this study? How to weigh the impact of associated injuries other than abdomen on the outcomes of laparotomy? Because the results of this study might give trauma surgeons more information about current profile of outcomes of trauma laparotomy over the world, so I have to figure out the problem of diversity of patient selection of the study. I thought the issue is also important to many readers of BMJ open.
---

REVIEWER	Sylvris, Amy Royal Melbourne Hospital Department of Surgery
REVIEW RETURNED	29-Jan-2024

GENERAL COMMENTS	This looks to be an interesting study that will certainly add to the current literature regarding trauma laparotomies. I wonder whether the authors would consider further data collection regarding indications for the trauma laparotomies. Given this is a multi-centre trial, it is likely that the clinical decision-
--

	making will be variable between centres and surgeons. It would be useful to know details such as the level of care at the hospital (eg whether it is a trauma centre, which would impact experience of the surgeon with trauma, etc) as well as the geographical location and resource availability (ICU beds, rurality, etc). Could you also please clarify whether you will include patients who undergo an initial trauma laparoscopy but subsequently are converted to laparotomy? Given that trauma laparoscopies are increasingly utilised, it would be more unique and up-to-date if your study were to provide data regarding outcomes for patients who require a conversion to open procedures.
--	--

VERSION 1 – AUTHOR RESPONSE

Reviewer: 1

Although the design of the study is great and the proposed problem about the outcomes of trauma laparotomy is important.

We thank the reviewer for these positive comments regarding the study.

I still have concerns about patient selection of the study. This study tries to analyze the outcomes of patients undergoing laparotomy for abdominal trauma either by penetrating or blunt mechanism, and the impact of higher incidence of associated injuries for blunt abdominal trauma might be a confounding factor. I know it's not easy to incorporate as many as 582 patients undergoing laparotomy for trauma in a 30-day period, but recruiting patients by different trauma mechanisms might be a problem for further analysis of data. Could the study designers give more explanations about the rationale about patient selection of this study? How to weigh the impact of associated injuries other than abdomen on the outcomes of laparotomy? Because the results of this study might give trauma surgeons more information about current profile of outcomes of trauma laparotomy over the world, so I have to figure out the problem of diversity of patient selection of the study. I thought the issue is also important to many readers of BMJ open.

Thank you for these comments.

Firstly, we are in agreement with the reviewer regarding burden on injury from blunt abdominal trauma, however we aim to quantify severity / burden on injury through calculation of the Injury Severity Score (ISS). The ISS is a validated and widely used metric for such use, therefore by adjusting for both mechanism of injury and ISS, we would hope that the issues surrounding burden of disease can be accounted for.

Furthermore, we have purposely kept the inclusion criteria of the study broad (i.e. all trauma laparotomy, any age, any mechanism) to ensure this best matches the study's objective. The GOAL-Trauma study aims to better understand trauma pathways and trauma care globally, using the trauma laparotomy as a surrogate procedure for this. Whilst important, the further details and specifics involved in trauma laparotomy patients (e.g. fluid resuscitation regimes, specific novel procedures) are less of the focus.

We hope that the outcomes for this study will identify key areas of improvement required in the trauma care pathways across the globe, alongside providing key epidemiological metrics to further support the strengthening of trauma systems worldwide.

Reviewer 2

This looks to be an interesting study that will certainly add to the current literature regarding trauma laparotomies.

We thank the reviewer for their kind feedback about the study; we are in agreement, in that we believe the GOAL-Trauma study will provide key data and metrics to help support the strengthening of trauma pathways and systems across the globe.

I wonder whether the authors would consider further data collection regarding indications for the trauma laparotomies. Given this is a multi-centre trial, it is likely that the clinical decision-making will be variable between centres and surgeons. It would be useful to know details such as the level of care at the hospital (eg whether it is a trauma centre, which would impact experience of the surgeon with trauma, etc) as well as the geographical location and resource availability (ICU beds, rurality, etc).

We agree with the reviewer's comments, awareness of the location, hospital type, funding etc. will be key to better understanding some of the outputs from the GOAL-Trauma study. As part of the study, each centre's local lead will be asked to complete a Centre Questionnaire, to obtain this exact information. This will provide a much richer data set and complement much of the quantitative data that the main study set will collect. We have edited the manuscript accordingly to better reflect this.

Could you also please clarify whether you will include patients who undergo an initial trauma laparoscopy but subsequently are converted to laparotomy? Given that trauma laparoscopies are increasingly utilised, it would be more unique and up-to-date if your study were to provide data regarding outcomes for patients who require a conversion to open procedures.

Thank you for raising this important point, one that was much discussed by our group during the development of the study protocol.

We are not including patients who have undergone laparoscopic procedure (including laparoscopic converted to open), alongside any robotic or interventional radiology procedure, to ensure the core data set provides the clearest picture on trauma provision and pathways globally. We are using the trauma laparotomy as using the trauma laparotomy as a surrogate procedure to map this care. Whilst important, the further details and specifics involved in trauma laparotomy patients (e.g. criteria for use of laparoscopy) are less of the focus.

Whilst we fully appreciate the reviewer's comment that trauma laparoscopy is becoming increasing used, this is only within a select group of countries worldwide. This study aims to map out the trauma care of patients worldwide, therefore inclusion of trauma laparoscopy would limit the GOAL-Trauma study's ability to address this.